



# CIOFC1.0: a Common Parallel Input/Output Framework Based on C-Coupler2.0

Xinzhu Yu[1], Li Liu[1,2], Chao Sun[1], Qingu Jiang[1,3], Biao Zhao[4,1], Zhiyuan Zhang[5], Hao Yu[1], Bin Wang[1,2,6]

[1] Ministry of Education Key Laboratory for Earth System Modeling, Department of Earth System Science, Tsinghua University, Beijing, China

[2] Southern Marine Science and Engineering Guangdong Laboratory (Zhuhai), China

[3] CMA Earth System Modeling and Prediction Centre (CEMC), China

[4] First Institute of Oceanography, and Key Laboratory of Marine Science and Numerical Modeling, Ministry of Natural
Resources, Qingdao, China

[5] Unit No. 91001 of PLA, Beijing, China

[6] State Key Laboratory of Numerical Modeling for Atmospheric Sciences and Geophysical Fluid Dynamics (LASG),
Institute of Atmospheric Physics, Chinese Academy of Sciences, Beijing, China

*Correspondence to*: Li Liu (liuli-cess@tsinghua.edu.cn)

**Abstract**

As Earth system modeling develops ever finer grid resolutions, the inputting and outputting (I/O) of the increasingly large data fields becomes a processing bottleneck. Many models developed in China, as well as the Community Coupler (C-Coupler), do not fully benefit from existing parallel I/O supports. This paper reports the design and implementation of a

Common parallel Input/Output Framework based on C-Coupler2.0 (CIOFC1.0). Parallelization by CIOFC1.0 can accelerate the I/O of large data fields. The framework also allows convenient specification by users of the I/O settings; e.g., the data fields for I/O, the time series of the data files for I/O, and the data grids in the files. The framework can also adaptively input data fields from a time-series dataset during model integration, automatically interpolate data when necessary, and output fields either periodically or irregularly. CIOFC1.0 demonstrates the cooperative development of an I/O framework and

coupler, and thus enables convenient and simultaneous use of a coupler and an I/O framework.

## 1 Introduction

Earth system models generally integrate component models of the atmosphere, ocean, land surface, and sea ice to support





climate change studies and provide seamless numerical predictions (Brunet et al., 2015). A coupler is a significant component or library in an Earth system model that effectively handles coupling among the component models. The various families of coupler include MCT (The Model Coupling Toolkit) (Larson et al., 2005), OASIS (Ocean Atmosphere Sea Ice Soil) (Redler et al., 2010; Valcke, 2013; Craig et al., 2017), CPL (The CESM Coupler) (Craig et al., 2005; Craig et al., 2012),

YAC (Yet Another Coupler) (Hanke et al., 2016), and C-Coupler (The Community Coupler) (Liu et al., 2014; Liu et al., 2018). There are also model frameworks with coupler capabilities, such as ESMF (Earth System Modeling Framework) (Valcke et al., 2012) and FMS (Flexible Modeling System) (Balaji et al., 2006). This paper focus on C-Coupler, a coupler family developed in and widely used in China (Li et al., 2020b; Lin et al., 2020; Shi et al, 2021; Ren et al., 2021; Wang et al., 2018; Zhao et al., 2017).

As models are under the development of finer grid resolutions, both they and the associated couplers are required to input and output (I/O) increasingly large data files, and I/O becomes a bottleneck in model simulations. More and more component models employ parallel I/O supports (e.g., MPI-IO (Message Passing Interface-I/O) and PnetCDF (Parallel Network Common Data Format); Li et al., 2003) to accelerate I/O; e.g., the atmosphere model CAM (Community Atmosphere Model) (Neale et al., 2010), GRAPES (Global/Regional Assimilation and Prediction System) (Zou et al., 2014) and a global cloud

resolving model (Palmer et al., 2011). Some common parallel I/O frameworks such as XIOS (XML Input/Output Server) (Yepes-Arbós et al., 2022) can be shared by various component models. Many models developed in China do not fully benefit from parallel I/O supports, and instead use sequential I/O for high-resolution integration; e.g., the GAMIL (grid-point atmospheric model of IAP LASG) atmosphere model (Li et al., 2013a; Li et al., 2013b; Li et al., 2020a) and C-Coupler.

To assist in model development in China, this paper reports the design and development of the Common parallel I/O

Framework based on C-Coupler2.0 (CIOFC1.0). The framework can benefit not only C-Coupler but also various component models. The remainder of this paper is organized as follows. Sections 2 and 3 respectively introduce the overall design and the implementation of CIOFC1.0. The framework is evaluated in Section 4, and a discussion and conclusions are provided in Section 5.

## 2 Overall design of CIOFC1.0

With the aim of aiding model development, we considered the following main requirements when designing the new I/O framework.

1)    The framework can obviously accelerate data input and output by employing parallel I/O supports, especially under a fine resolution.





2) The framework should adaptively input time-series data fields from a set of data files and automatically conduct time interpolation when required.

3) The framework should facilitate outputting of data either periodically or irregularly. Periodic output is a traditional requirement, but atmospheric models such as GRAPES (Zhang and Shen, 2008), which is now used for national operational weather forecasting in China, does not use a uniform period for data output: it generally outputs data for three hour intervals in the first five model days and then for six hour intervals in the remaining model days. The atmospheric chemistry model GEOS-Chem (Long et al., 2015) enables users to specify a set of specific model dates for outputting model data.

4) The framework should automatically conduct spatial interpolation in parallel when a data field is on different grids in the model and in the data files. Users generally expect fields in data files to be on regular grids (particularly longitude–latitude grids), while models increasingly employ irregular grids. For example, the atmospheric model of FV3 (Finite Volume Cubed SphereFinite Volume Cubed Sphere) (Putman and Lin, 2007) and MCV (Multimoment Constrained Finite-Volume Model) (Li et al., 2013c; Chen et al., 2014; Tang et al., 2021) use cubed-sphere grids, the atmospheric model MPAS-A (Heinzeller et al., 2016) generally uses unstructured grids generated by triangulation (Jacobsen et al., 2013; Yang et al., 2019), and many ocean models (e.g., POP (Parallel Ocean Program) (Smith et al., 2010), LICOM (LASG/IAP climate system ocean model) (Lin et al., 2016; Liu et al., 2012) and MOM (Modular Ocean Model) (Griffies, 2012)) use tripolar grids. Spatial interpolation therefore becomes necessary when inputting and outputting data fields for models.

5) The framework should facilitate flexible and convenient specification of the I/O settings; e.g., the data fields to be input or output, the time series of the input data files or of the output data, and the data grids in files (called *file grids* hereafter).

We designed the main architecture of CIOFC1.0 considering the above requirements. Fig. 1 shows that it comprises the following set of modules: input time series manager, output time series manager, spatial data interpolation manager, parallel I/O operation, output driving procedure, input driving procedure, and I/O configuration manager. These modules enable convenient use of CIOFC1.0 via a set of Application Programming Interfaces (APIs) and XML configuration files.

1) The input time series manager handles time-series information of data fields in a set of input data files. It enables users to flexibly specify rules for time mapping between data files and models. Given a model time, it determines whether to



conduct time interpolation and whether to input the field values at a corresponding time point in a corresponding data file.

2) The output time series manager enables a component model to output model data periodically or irregularly. It determines whether a component model should output model data at the current model time.

3) The spatial data interpolation manager manages the model grid and the file grid for each field. It automatically conducts parallel spatial data interpolation when the model grid and file grid of a field are different.

4) The parallel I/O operation employs parallel I/O supports to write each field into a specific data file and read in data values from a specific data file in parallel.

5) The I/O configuration manager enables users to flexibly specify I/O configurations for a component model via XML

formatted configuration files; e.g., configurations for the input/output time series, file grids, and each input/output field.

6) The input driving procedure and output driving procedure respectively organize the procedure for inputting and outputting field values based on other modules.

**3 Implementation of CIOFC1.0**

This section introduces details of each module of CIOFC1.0.

**3.1 Implementation of the I/O configuration manager**

The I/O configuration manager formats a set of XML files for the configuration of the file grids, input/output time series, and each input/output field.

**3.1.1 Configuration of file grids**

Specifying file grids via model codes can be inconvenient because users generally must modify and then recompile the

model codes when changing file grids in different simulations. The I/O configuration manager therefore provides XML configurations for file grids as a better option for users.

A challenge here is making the configurations as widely compatible as possible for various grids. C-Coupler can handle various kinds of horizontal grid, support several kinds of vertical coordinates, and represent a 3-D grid in a "2-D + 1D" or "1D + 2-D" manner. We therefore design configurations for horizontal grids, vertical coordinates, and 3-D grids.



### 3.1.1.1 Configurations for horizontal grids

Considering that file grids are usually regular longitude–latitude grids, we simplify the specification of a regular longitude–latitude grid into several parameters in the XML configuration file (e.g., lines 2 to 8 in Fig. 2): the parameters "*min_lon*", "*max_lon*", "*min_lat*", and "*max_lat*" specify the domain of the grid, while the parameters "*num_lons*" and

5 "*num_lats*" specify the grid size. Thus, the 2-D coordinate values of a longitude–latitude grid can be calculated automatically by CIOFC1.0.

As it is difficult for CIOFC1.0 to automatically calculate coordinate values for an irregular grid, a specification rule is designed to instruct CIOFC1.0 to read in coordinate values from a file (e.g., lines 9 to 17 in Fig. 2), where the XML attribute "*file_name*" specifies the file and the attributes "*center_lon*" and "*center_lat*" specify the fields of the center coordinate

values in the file. The file fields for vertex coordinate values, mask, and area of each grid cell can be further specified via the XML configuration file (not shown in Fig. 2).

The specification of a file grid can be further simplified into a unique file name (e.g., lines 18 to 22 in Fig. 2) when it corresponds to a NetCDF file that matches C-Coupler's default grid data file format (e.g., Fig. 3).

In an XML configuration file, users can specify several horizontal grids that are identified by different grid names.

### 3.1.1.2 Configurations for vertical coordinates

C-Coupler currently supports three kinds of vertical coordinates: Z, SIGMA, and HYBRID coordinates. We therefore design a specification for each type (lines 2 to 7, lines 8 to 13, and lines 14 to 21 in Fig. 4 give examples of each of the three types, respectively). Values of the vertical coordinate can be obtained from fields of a file (lines 14 to 21 in Fig. 4), specified by an

20 explicit array of values (lines 2 and 7 in Fig. 4), or generated automatically according to the specified boundaries under a descending or ascending order (lines 8 to 13 in Fig. 4).

In the XML configuration file, users can also specify multiple vertical coordinates that are identified by different grid names.

### 3.1.1.3 Configurations for 3-D grids

In response to the "2-D + 1D" and "1D + 2-D" representations of a 3-D grid in C-Coupler, CIOFC1.0 enables users to specify the horizontal and vertical sub grids of a 3-D grid in the XML configuration file (e.g., lines 2 to 8 and lines 9 to 12 in Fig. 5). Given a 3-D grid with a vertical sub-grid of SIGMA or HYBRID coordinate, information about the surface field should be specified. A surface field can be specified as a field read from a data file (lines 17 to 28 in Fig. 5); it can



alternatively be specified as "external", meaning that values of the surface field originate from the component model when outputting a field. In a possible special case, users may want a file grid to use the same horizontal or vertical sub-grid as the model. A special horizontal grid name "*handler_output_H2D_grid*" (lines 9 to 12 in Fig. 5) and a special vertical sub-grid name "*handler_output_V1D_grid*" (lines 13 to 16 in Fig. 5) are employed for such a case.

### 3.1.2 Configurations for outputting fields

Considering that users may want to output different fields with different settings (e.g., different time series, different file grids, and different data types), CIOFC1.0 enable users to specify a specific configuration for a group of fields (e.g., lines 8 to 12 and lines 18 to 20 in Fig. 6). For a field group, users can specify common output settings (line 2 in Fig. 6); e.g., frequency of creating a new files (corresponding to the XML attributes "*file_freq_count*" and "*file_freq_unit*"), default data types, output time series ("…" in line 5, will be further discussed in Section 3.4), time format in data files, outputting instantaneous values or time-averaged values, and file grids. A field in a group can also have its own specific settings (line 9 in Fig. 6). Users can make a field use the same name in both the model and data files (line 11 in Fig. 6) or make data files use a new field name (the corresponding XML attribute "*name_in_file*" is specified; line 9 and 10 in Fig. 6).

How to specify the output time series will be further introduced in Section 3.2.

### 3.1.3 Configurations for inputting time-series data fields

Considering that the time-series fields in data files can be shared by different component models and different simulation settings, the configurations for inputting time-series data fields are divided into two parts: information about a time-series dataset, and how to bridge a dataset and a component model.

For specifying the data files included in a dataset, CIOFC1.0 enables a dataset to have a unique data file (e.g., line 3 in Fig. 7(a)) or consist of a group of data files whose names are different only in terms of time (line 3 in Fig. 7(b), where the unique character "*" in the common file name will be automatically replaced by the time string under the specified time format when determining the name of each input data file). The time points in a time series can be obtained from the name of the data files (e.g., corresponding to the character "*" in line 3 in Fig. 7(b)) or from a set of time fields in the data files (lines 4 to 7 in Fig. 7(a)) with a specified type of time points (i.e., "start", "middle", or "end"; line 4 in Fig. 7(b)). For example, given a





time format "MMDDHH" (representing month, day, and hour), "start", "middle" and "end" means the $0^{th}$, $1800^{th}$ and $3600^{th}$ second in each hour. The fields provided by a group of data files, as well as the grid of each field, should be listed explicitly (lines 21 to 23 in Fig. 7(a)). Each configuration file contains only one dataset with a unique dataset name as the keyword (e.g., the dataset named "dataset1" corresponding to line 1 in Fig. 7(a)).

A component model can call the corresponding API (Section 3.7) to input fields from a time-series dataset based on an input instance with a unique name (corresponding to the XML attribute "*instance_name*" in line 2 in Fig. 8). As the same field may have different names in a dataset and a component model, specification of name mapping for each field is enabled (lines 7 to 10 in Fig. 8). The time range of a dataset is generally fixed, while users may want flexible use of a dataset in different

simulations. For example, a pre-industrial control simulation of a coupled model corresponding to the Coupled Model Intercomparison Projects always uses forcing data for the year 1850. Scientists may want to use different time ranges for atmosphere forcing data with the same time range as emissions forcing data in atmospheric chemistry simulations, to study the impact of climate variation on air pollution. We therefore enable users to specify mapping rules between the dataset time and model time (lines 3 to 6 in Fig. 8); i.e., uses can set a fixed offset between the model time and data time, a fixed period

for periodically using time-series data, or even a hybrid combination of the two.

To further improve the commonality in specifying the input time series manager, as well as in the whole CIOFC1.0 system, various time formats (Table 1) are supported.

**3.2 Implementation of the spatial data interpolation manager**

When a field has a file grid different from its model grid, spatial data interpolation for this field is required. As C-Coupler2 already implements horizontal (2-D) interpolation and 3-D interpolation in a "2-D + 1D" manner, the spatial data interpolation manager directly employs this functionality (for further details of this functionality, see Liu et al. (2018)). Specifically, the spatial data interpolation manager will generate the file grid of each field according to the specifications in

the XML configuration file, and then generate remapping weights and conduct data interpolation as required. Different fields with the same model grid and the same file grid can share the same remapping weights.



With such an implementation, CIOFC1.0 enables a model with an unstructured horizontal grid (e.g., a cubed-sphere grid, a non-quadrilateral grid, or even a grid generated by triangulation; Yang et al., 2019) to output fields to a regular longitude–latitude grid in data files. CIOFC1.0 also enables a model with SIGMA or HYBRID vertical coordinates to output fields to Z vertical coordinates in data files, as C-Coupler2 supports the dynamic interpolation between two 3-D grids, either

of which can calculate variable vertical coordinate values following the change of the corresponding surface field in time integration.

We identified a limitation of C-Coupler2 when applying CIOFC1.0 to the atmosphere model MCV that outputs 3-D fields to regular longitude–latitude grids and barometric surfaces in data files. Although C-Coupler2 can handle data interpolation between a regular longitude–latitude grid and the cubed-sphere grid used by MCV, it cannot handle vertical interpolation to

barometric surfaces, because MCV uses SIGMA vertical coordinates of height but not barometric pressure. We therefore developed a new API *CCPL_set_3D_grid_3D_vertical_coord_field* and the corresponding functionalities to dynamically use 3-D barometric pressure values diagnostically calculated by MCV during the vertical interpolation to barometric surfaces.

**3.3 Implementation of the parallel I/O operation**

Considering that models for Earth system modeling generally access data files in NetCDF format and that C-Coupler implements NetCDF as the default data file format, we use PnetCDF to develop the parallel I/O operation. PnetCDF enables a group of processes to cooperatively output/input multi-dimensional values of a field to/from a data file, where calling the corresponding PnetCDF APIs (Fig. 9) enables a process to output/input the field values in a subdomain or a subspace

specified via the API argument arrays "*starts*" and "*counts*". The subdomains or subspaces associated with each process are generally determined by the parallel decomposition. For example, as determined by the regular 2-D decomposition in Fig. 10(a), each process is associated with one subdomain, while the round-robin-based decomposition in Fig. 10(b) determines that each process is associated with multiple subdomains. To enable a process to output/input the values in multiple subdomains or subspaces, a straightforward implementation is to call the asynchronous PnetCDF APIs (e.g.,

*ncmpi_iput_var\** and *ncmpi_iget_var\**) multiple times before the calling the API *ncmpi_wait_all*.

Although PnetCDF can rearrange data values in subdomains or subspaces among processes to improve parallel I/O performance (Thakur et al., 1999), we do not recommend the above implementation as it does not fully benefit from

C-Coupler2, especially the functionality of data rearrangement. Specifically, C-Coupler2 takes two main steps for data rearrangement: generating a routing network among processes at the initialization stage, and MPI communications following the routing network at each time of data rearrangement. Optimizations of these two steps have been investigated (Yu et al., 2020; Zhang et al., 2016). For a better implementation, a group of I/O processes that are a subset of model processes call

PnetCDF APIs, and a regular parallel decomposition on all I/O processes (called I/O decomposition hereafter) is generated. Such an implementation makes each I/O process associated with a unique subdomain (or subspace) with an averaged number of grid points, and enables C-Coupler2 to work for the data rearrangement between model processes and I/O processes.

A parallel I/O operation implements the following steps to output a field:

1)   When the I/O decomposition of this field has not been generated, generate the I/O decomposition and the corresponding routing network between model processes and I/O processes.

2)   Rearrange the field values from model processes to I/O processes.

3)   Call PnetCDF APIs to write field values into a file in parallel.

A parallel I/O operation implements the following steps to input a field:

1)   Generate the I/O decomposition and the corresponding routing network when required (the same as the first step for outputting a field).

2)   Call PnetCDF APIs to read in field values from a file in parallel.

3)   Rearrange the field values from I/O processes to model processes.

The number of I/O processes can affect the efficiency of parallel I/O operation. It is difficult to automatically determine an optimal number of I/O processes, because it depends greatly on the number of model processes and the hardware/system supports in a high-performance computer. Currently, we enable users to specify a global number of I/O processes via a configuration file (the parameter "*max_num_pio_proc*" in Fig. 11).

### 3.4 Implementation of the output time series manager

C-Coupler2 can simply describe the timing of a periodic time series using the period unit and period count. Given a restart timer with unit *month* and count *5*, the model simulation outputs a restart data file every five model months. However,

irregular time series have not previously been supported by C-Coupler2. A straightforward solution to describe an irregular time series is to enumerate all time points in the series. We do not recommend this, because it will be inconvenient for long simulations with many time points. A better solution relies on irregular time series generally consisting of several parts with regular periods. For example, the weather forecasting model GRAPES mentioned in Section 2 generally uses two periods for

outputting: every three hours for the first five model days and every six hours for the remaining model days. The output time series manager should therefore enable users to specify a time series with multiple non-overlapping time slots, most of which have a uniform period. Any time slot with no uniform period should be handled by the output time series manager, enabling users to enumerate all the time points in that slot.

To support various kinds of time series in a convenient manner, the output time series manager employs the terms of period, time slot, and time point, supports flexible combination among them, and enables users to specify an output time series via the XML configuration file. Time slots can be nested in a period, so a period can contain multiple time slots. A period can also be nested in a time slot, which means that a time slot can contain a periodic time series; a period or a time slot can contain multiple time points.

In the example XML specification in Fig. 12(a) (lines 3 to 12), the outermost level is a period of every day (line 4). Two groups of time slots are nested in this period (lines 5 to 10). The time slots of the 1st to 2nd, 9th to 10th, and 17th to 18th hour in the first group nest the same period of every hour (line 5 in Fig. 12(a)). The time slots of the 5th to 6th, 13th to 14th, and 21st to 22nd hour in the second group nest the same period of every two hours (line 8 in Fig. 12(a)). The period in each time slot

group further includes time points (lines 6 and 9). This example specification determines a complex irregular time series (all time points in one day are shown in Fig. 12(b)).

Fig. 14 shows the XML specification corresponding to the specific output settings of GEOS-Chem in Fig. 13. The outermost level is the period of every year (line 2 in Fig. 14). There are three groups of time slots nested in this yearly period (lines 3 to

9). The first group (lines 3 to 5) means that the model outputs data every three hours in the first day of each month from January to May and from August to December. The second group (line 6) means that the model outputs data every three hours in every day of June. The third group (lines 7 to 9) means that the model outputs data every three hours in the first and second days of July. Similarly, the output time series of GRAPES mentioned above can be easily specified, as shown in Fig. 15 (lines 1 to 4), where the XML attribute value "*ndays*" means the number of model days since the start of the simulation.

The output time series manager extends the timer functionality of C-Coupler2—which is only compatible with periodic time series—with a tree of timers to keep the nesting relationship among periods, time slots, and time points. The timer tree of an output time series is initialized when loading the corresponding specifications from the XML configuration file, where the

correctness of the nesting relationship will be checked. For example, the specifications in Fig. 16 are incorrect, because the minimum time unit of the period at the outermost level (line 2) is *hour*, while the minimum time unit of the second level (line 3) is *day*, which is longer than *hour*. When the model attempts to output a set of fields, it checks whether the corresponding timer tree is on at the current model time.

Each node in a timer tree corresponds to the timer of a period, a set of time slots, or a set of time points; a node may have multiple children. A recursive procedure is implemented to handle the tree structure. Given a tree node, the procedure first checks whether the corresponding timer is on, and then checks each child of the tree node recursively. A tree node is on at the current model time only when its corresponding timer is on, and it has no child or at least one child node is on. A timer of time slots/points is on when the current model time matches one time slot/point.

For example, Fig. 17 shows the timer tree corresponding to the output time series specified in Fig. 12(a). The root tree node ("*node1*") corresponds to the outermost periodic timer. Given a model time of date 20060101 and second 33600, it matches the period of every day in "*node1*", the time slot 9–10 hour in "*node2*", the period of every hour in "*node4*", and the time point of 1200 s in "*node6*". So, the timer tree is on at this model time. Given a model time of date 20060101 and second

19800, it matches the period of every day in "*node1*", the time slot 5–6 hour in "*node3*", the period of every two hours in "*node7*", but does not match the time points of 2400 and 4800 s in "*node8*" (the time point corresponding to the given model time is 1800 s). As a result, the timer tree is not on at this model time.

**3.5 Implementation of the output driving procedure**

The output driving procedure enables a model to apply multiple output handlers corresponding to different groups of fields or different output configurations. Specifically, the API *CCPL_register_configurable_output_handler* is designed for applying an output handler with complex configurations specified in the XML file, while the API





*CCPL_register_normal_output_handler* is also designed for use with an output handler with simple configurations specified in the argument list but not in the XML file. An output handler can be executed implicitly when the model is advancing model time, or executed explicitly through calling the API *CCPL_handle_normal_explicit_output*.

The implementation of the output driving procedure will be further examined for each API.

### 3.5.1 Implementation corresponding to the API *CCPL_register_configurable_output_handler*

The API *CCPL_register_configurable_output_handler* provides an argument list for flexibility (Fig. 18). Each output handler has a unique name (corresponding to the argument "*handler_name*") and works for a group of fields (corresponding

to the arguments "*num_field_instances*" and "*field_instance_ids*") of the same component model that has already been registered to C-Coupler2. The XML configuration file corresponding to an output handler can be specified via the argument "*configuration_name*". Such an implementation enables a component model to use different configurations to output the same group of fields or use the same configuration to output different groups of fields. The execution of an output handler can be either implicit or explicit (corresponding to the argument "*implicit_or_explicit*"). Users can specify a default file grid

(corresponding to the optional argument "*output_grid_id*") that has already been registered to C-Coupler2, and thus the default file grid will be used for a field when the file grid of this field is not specified in the configuration file. The arguments "*handler_output_H2D_grid_id*" and "*handler_output_V1D_grid_id*" correspond to the special sub grids of "*handler_output_H2D_grid*" and "*handler_output_V1D_grid*" for configuring 3-D grids in a configuration file (Section 3.1.3). An output handler can be executed at every model time step or under a specified periodic timer (corresponding to the

optional argument "*sampling_timer_id*"). For example, given that a component model with a time step of 60 s outputs time-averaged values of a group of fields every three model hours, and that users specify a periodic timer of every hour, the values outputted at the third model hour are averaged from the first three model hours, but not from every model step.

The API *CCPL_register_configurable_output_handler* implements the following main steps to initialize an output handler.

1)  Employ the I/O configuration manager to load output configurations from the corresponding XML file. Users will be notified if any error in the XML file is detected.

2)  Employ the spatial data interpolation manager to generate all file grids determined by the output configurations.

3)  Employ the output time series manager to generate the timer tree of each output time series (called output timer





hereafter) determined by the output configurations, and generate the periodical timer for creating new data files (called file timer hereafter).

4) Determine the output configuration corresponding to each field (e.g., file grid, output time series, file data type, and outputting time-averaged values or instantaneous values).

5) For each file grid, generate a parallel I/O operation as well as the corresponding I/O decomposition, and then allocate the corresponding I/O field instance.

6) Employ the coupling generator of C-Coupler2 to generate the coupling procedure from a model field to the corresponding I/O field instance. A coupling procedure can include a group of operations such as data transfer, data interpolation, data type transformation, and data averaging (for further details, see Liu et al. (2018)), while the spatial data interpolation manager will generate remapping weights for the data interpolation when necessary. Multiple fields that share the same model grid and the same file grid can share the same I/O field instance and the same coupling procedure, to save memory consumption.

### 3.5.2 Implementation corresponding to the API *CCPL_register_normal_output_handler*

Using the argument list shown in Fig. 19, the API *CCPL_register_normal_output_handler* works similarly to the API *CCPL_register_configurable_output_handler*, but does not rely on the configurations in the XML file. It can output a group of fields to the same file grid (corresponding to the optional argument "*output_grid_id*") that has been previously registered to C-Coupler2. Users can specify a periodic timer for outputting fields (corresponding to the optional argument "*output_timer_id*") and a periodic timer for generating new data files (corresponding to the optional argument "*file_timer_id*"). Users can also specify to output time-averaged or instantaneous values of all fields (corresponding to the optional arguments "*inst_or_aver*"), and specify the data types of fields in data files (corresponding to the optional arguments "*float_datatype*" and "*integer_datatype*").

The API *CCPL_register_normal_output_handler* uses a flowchart similar to the API *CCPL_register_configurable_output_handler*, but without the first and second main steps in Section 3.5.1.

### 3.5.3 Implementation corresponding to the API *CCPL_handle_normal_explicit_output*

The API *CCPL_handle_normal_explicit_output* explicitly executes an output handler that has been applied by calling the



API *CCPL_register_configurable_output_handler* or *CCPL_register_normal_output_handler*. It is generally called at each model step, but actually runs when the sampling timer of the output handler (corresponding to the argument "*sampling_timer_id*") is not specified or when the sampling timer is on at the current model time. When it does run, it first generates a new data file when required, and then iterates on each field. For a field, it conducts time-averaging if required,

conducts data interpolation if required, rearranges the field values on model processes to the I/O processes, and finally outputs the field values on the I/O processes to the data file when the output timer of the field is on. The API *CCPL_handle_normal_explicit_output* enables users to specify a special argument of "*bypass_timer*" to disable all timers, which means that the instantaneous values of each field will be outputted to a newly created data file.

The implicit execution of an output handler works similarly to the API *CCPL_handle_normal_explicit_output*.

### 3.6 Implementation of the input time series manager

The input time series manager manages the time-series information of each input dataset. All time points of an input dataset,

as well as the data file and time index corresponding to each time point, are recorded after parsing the corresponding XML configuration file.

Given the current model time, the input time series manager follows the specified mapping rules between the dataset time and the model time, and looks up the corresponding time points of an input dataset; i.e., one time point that matches the

current model time or two adjacent time points where the current model time is in the interval between them.

Corresponding to the configuration of a daily input dataset in Fig. 21 (lines 3 to 6), Table 2(a) shows the time fields in each of four data files. After parsing this configuration, all daily time points are loaded from these four data files and then extended with a second number (i.e., second 43200) according to the specified type of the time points ("middle" in Fig. 21),

as shown in Table 2(b). Lines 3 to 6 in Fig. 22 specify a time mapping rule hybrid with a period of 20 days and an offset of 5 days. Such a rule determines that the first model day corresponds to the sixth day in the period. Given a simulation starting from a model time of date 20211101 and second 43200, only one time point (i.e., 2005010643200) in the input dataset corresponds to the start time, while there are two time points (i.e., 2005012043200 and 2005010143200) corresponding to





---

the model time of date 20211115 and second 46800.

### 3.7 Implementation of the input driving procedure

To enable a model to iteratively input a group of fields from time-series datasets in time integration, the input driving procedure provides an API *CCPL_register_input_handler* for applying an input handler and provides an API *CCPL_execute_input_handler* for iteratively executing the input handler. Considering that the initial fields of a model are not generally time-series data, an API *CCPL_readin_field_from_dataFile* reads in values of a field from a specific data file. Implementation of the input driving procedure is further introduced in the following discussion on each API.

#### 3.7.1 Implementation corresponding to the API *CCPL_register_input_handler*

The API *CCPL_register_input_handler* provides an argument list like that shown in Fig. 23. Similar to the output handler (Section 3.5), each input handler also has a unique name (corresponding to the argument "*handler_name*"), and works for a group of fields (corresponding to the arguments "*num_field_instances*" and "*field_instance_ids*") of the same component model that has already been registered to C-Coupler2. The configurations for inputting field values from time-series datasets (called input configurations hereafter) should be specified (via the argument "*config_input_instance_name*"). The input

handler can also be executed under a specified periodic timer (corresponding to the optional argument "*input_timer_id*"). Considering that users may require flexibility to change the source of each boundary field of a component model (i.e., the source of a boundary field can be either another component model or a time-series dataset) in different simulations of a coupled model, two optional arguments "*necessity*" and "*field_connected_status*" are provided. The array "*necessity*"

enables users to specify that each field is necessary or optional (i.e., a necessary field must be input from a dataset and users must specify the input configurations of each necessary field, while users can disable the source of an optional field from a dataset through not specifying the input configurations of the field), while an element in the array "*field_connected_status*" notifies whether the input configurations of the corresponding field have been specified. Thus, users can flexibly enable or disable the source of an optional field from a dataset through modifying the corresponding XML configuration file without

modifying the model codes.

The API *CCPL_register_input_handler* implements the following steps to initialize an input handler.



1) Employ the I/O configuration manager to load input configurations from the corresponding XML file. Users will be notified if any error in the XML file is detected.

2) Employ the spatial data interpolation manager to generate all file grids determined by the output configurations.

3) Determine the input configuration corresponding to each field, file grid, data set, data type, etc.

4) For each file grid, generate a parallel I/O operation as well as the corresponding I/O decomposition, and then allocate the corresponding I/O field instance.

5) Employ the input time series manager to manage the time-series information of each field and the time mapping rule.

6) Generate the coupling procedure from an I/O field instance to the corresponding model field. This step is similar to the last step of the *CCPL_register_configurable_output_handler*.

### 3.7.2 Implementation corresponding to the API *CCPL_execute_input_handler*

The API *CCPL_execute_input_handler* executes an input handler that has been applied via an API call of *CCPL_register_input_handler*. It can be called at any model time, and can actually run when the input timer of the input handler is specified and is on at the current model time. When it actually runs, it iterates on each field that can be input from the datasets. For a field, it employs the input time series manager to determine the dataset time points corresponding to the current model time, employs the corresponding parallel I/O operation to read in field values from the dataset corresponding to each time point, rearranges the field values on the I/O processes to model processes, conducts spatial data interpolation operations if required, and finally conducts time interpolation if required.

Here we consider an example with the input configurations in Fig. 21 and Fig. 22. When inputting the model field "*tskin_in*" at the model time of date 20211101 and second 43200, there is only one dataset time point (i.e., 2005010643200) corresponding to this model time, and thus the field values corresponding to the first time point in the file "*file_name.20050106.nc*" will be input, and no time interpolation will be conducted. When inputting the model field "*tskin_in*" at the model time of date 20211101 and second 46800, there are two dataset time points (i.e., 2005010643200 and 2005010743200) corresponding to this model time, and thus the field values corresponding to the first and second time points in the file "*file_name.20050106.nc*" will be inputted, and time interpolation will be conducted.

To reduce I/O overhead, the input driving procedure will cache the field values that have been recently input from datasets.

For example, the field values corresponding to the dataset time point 2005010643200 are cached and will not be input again at the model time of date 20211101 and second 46800.

### 3.7.3 Implementation corresponding to the API *CCPL_readin_field_from_dataFile*

The API *CCPL_readin_field_from_dataFile* works for only one model field (corresponding to the argument "*field_instance_id*" in Fig. 25) and reads in the values of this field from a specific data file (corresponding to the argument "*data_file_name*"). It enables a field to have a different name (which can be specified via the argument "*field_name_in_file*") or a different grid (which can be specified via the argument "*grid_id_in_file*") in the data file. Users can further set maximum and minimum boundaries (corresponding to the arguments "*value_max_bound*" and "*value_min_bound*") to guarantee that no abnormal values out of the boundaries have been read in.

The functionality of the API *CCPL_readin_field_from_dataFile* can be achieved through combining the functionalities of the APIs *CCPL_register_input_handler* and *CCPL_execute_input_handler* together for a special case; i.e., only one field to be input without time information.

### 4 Evaluations of CIOFC1.0

This section evaluates CIOFC1.0 empirically in terms of functionality and performance. The framework was integrated into real models (GAMIL and MCV); we further developed a test model to enable evaluation of the framework through systematic adjustment of the settings.

All test cases were run on the High-Performance Computing System (HPCS) of the Earth system numerical simulator (http://earthlab.iap.ac.cn/en/). The HPCS has approximately 100,000 CPU cores (X86; running at 2.0 GHz) and 80 PB of parallel storage capacity. Each computing node includes 64 CPU cores and 256 GB memory, and all of the nodes are connected by a network system with a maximum communication bandwidth of 100 Gbps. All codes are compiled by an Intel Fortran and C++ compiler (version 17.0.5) at the O2 optimization level using an OpenMPI library.

### 4.1 Test model

Our development of the test model for evaluating CIOFC1.0 focused only on the capabilities for I/O fields, and numerical calculations in real models were neglected. The test model is capable of I/O of 2-D and 3-D fields based on CIOFC1.0 and offers two kinds of horizontal grid (i.e., longitude–latitude and cubed-sphere) and two types of parallel decomposition (i.e., regular 2-D and round-robin) for selection. The test model further enables flexible setting of the number of fields, the size of the horizontal grid, the number of vertical levels (the Z coordinate is used), and the number of model processes. It also allows to advance integration with a specified time step.

### 4.2 Functionality of CIOFC1.0

Functionality evaluation considered three categories: correctness of the I/O functionalities, correctness of spatial interpolation, and correctness of I/O time management.

#### 4.2.1 Correctness of the input and output functionalities

We combined the data input and output functionalities when evaluating their correctness; i.e., the test model inputs a set of fields from an existing data file and then outputs these fields into a new data file. We made the test model, the existing data file, and the new data file use the same grids to avoid any spatial data interpolation and associated interpolation error. Therefore, the existing data file and the new data file should be the same when the input and output functionalities are correct. Specifically, we prepared four input data files on cubed-sphere grids under different resolutions and four more on longitude–latitude grids, and finally obtained 16 test cases based on each input data file undergoing the two types of parallel decomposition supported by the test model. The results for each case show the correctness of the I/O functionalities.

CIOFC1.0 enables flexible user-specified setting of the number of I/O processes. A change of the number of I/O processes should not introduce any change to the fields' input or the output data file. Therefore, we further checked the correctness of the I/O functionalities by changing the number of I/O processes. Specifically, we used the 16 test cases above and used three different numbers of I/O processes to run each test case. The results show that CIOFC1.0 produces the same results under different numbers of I/O processes.

#### 4.2.2 Correctness of spatial interpolation

Spatial interpolation is an existing functionality of C-Coupler2. As its correctness has already been evaluated (cf. Figs 11 and

12 in Liu et al., 2018), here we only confirm that it works correctly for CIOFC1.0 if the file grid is different from the model grid in data input or output. Specifically, we reproduced the coupling of the fields temperature and zonal wind speed from GAMIL to GEOS-Chem in Liu et al. (2018), and further made GAMIL output these fields to the grid of GEOS-Chem based on CIOFC1.0. As the fields in the output file were the same as those obtained by GEOS-Chem through coupling, spatial

interpolation by CIOFC1.0 is shown to work correctly.

Dynamic 3-D interpolation is a capability that has been further advanced for CIOFC1.0: the new API *CCPL_set_3D_grid_3D_vertical_coord_field* and the corresponding functionalities enable the 3-D interpolation to use vertical coordinate values dynamically calculated by a model. We further evaluated the correctness of this new enhancement

based on MCV—which has an internal code for remapping fields to a regular 3-D grid (consisting of a regular longitude–latitude grid and barometric surfaces)—before outputting the fields. Specifically, we made MCV use CIOFC1.0 to output fields to the same regular 3-D grid. The data files output by MCV internal code and CIOFC1.0 were consistent, demonstrating that the new advancement of dynamic 3-D interpolation has been implemented correctly for CIOFC1.0.

### 4.2.3 Correctness of I/O time management

Using the test model, we evaluated the correctness of I/O time management for data input and output respectively. Specifically, we designed a set of different aperiodical output time series. We specified each output time series via the configuration file, ran the test model to get a set of output data files, and then checked whether the time series in these output data files was correct. Next, we made the test model input the aperiodical dataset from each set of output data files, and then checked whether time interpolation was conducted correctly (i.e., given each model time, whether CIOFC1.0 used the

correct time points in the input time series for time interpolation). The I/O time management functionality was demonstrated to work correctly in all test cases.

### 4.3 Performance of CIOFC1.0

This subsection evaluates the performance of CIOFC1.0 when inputting and outputting a field based on the test model. A fine resolution (5,000,000 horizontal grid points and 80 vertical levels) and a coarse resolution (50,000 grid points and 30

vertical levels) were used. Around 10,000 processor cores ran the test model at fine resolution, and around 500 cores were used for the coarse resolution. Each value in the data files is a single-precision floating point (4 bytes for each value). Each test case was run three times; average performance data are shown here.

We first evaluated the performance of data output. Fig. 26 (a) shows the time for outputting a 3-D field (1.6 GB) at the fine

resolution as the number of I/O processes increased from 1 to 8192. Both the data rearrangement time and the I/O time

decreased quickly as the number of I/O processes increased to 15. The data rearrangement time can be neglected when more

I/O processes are used, and the I/O time increased as the number of I/O processes increased above 128. Output time was

very low for 15–128 I/O processes. Fig. 26 (b) shows the time for outputting a 2-D field (20 MB) at the fine resolution. The

data rearrangement time can be neglected and minimal output times were achieved using 4–15 I/O processes. Fig. 27 (a)

shows the time for outputting a 3-D field (6 MB) at the coarse resolution for increasing numbers of I/O processes from 1 to

512; the output time was very low when using no more than 15 I/O processes. Fig. 27 (b) shows minimal output times for

outputting a 2-D field (200 KB) when using no more than four I/O processes.

We next evaluated the performance for data input. Fig. 28 (a) shows the time for inputting a 3-D field at the fine resolution;

minimal time was needed when using 32–128 I/O processes. According to Fig. 28 (b), very low time can be achieved with no

more than 128 processes when inputting a 2-D field. Inputting a 2-D or 3-D field at the coarse resolution required very little

time when using no more than four I/O processes (Fig. 29).

The above results show that although a small field does not benefit from parallel I/O, CIOFC1.0 can accelerate I/O data

when the field size in data files is large: e.g., compared with serial I/O (corresponding to using one I/O process in Figures 26

to 29), speed can be increased by between 5 and 34 times when respectively outputting and inputting a 3-D field at the fine

resolution.

**5 Conclusions and discussion**

In this paper, we propose a new common, flexible and efficient parallel I/O framework for Earth system modeling based on

C-Coupler2.0. CIOFC1.0 can handle data I/O in parallel, and provides a configuration file format that enables users to

conveniently change the I/O configurations (e.g., the fields to be output or input, the file grid, and the time series of the data

in files). It can automatically interpolate data when required: i.e., it can perform spatial data interpolation when the file grid

is different from the model grid, and also time interpolation in data input. It enables a model to output data with an aperiodical time series that can be conveniently specified via the configuration file. Empirical evaluation showed that CIOFC1.0 can accelerate data I/O when the field size is large.

A significant advancement achieved by CIOFC1.0 is the cooperative development of an I/O framework and coupler. CIOFC1.0 benefits greatly from the spatial data interpolation functionality of C-Coupler2.0. It needs few new APIs, because it can utilize the existing APIs of C-Coupler2.0 for registering model fields, grids, and parallel decompositions to C-Coupler2.0. As a result, a C-Coupler2.0 user can easily learn how to use CIOFC1.0 or conveniently use a coupler and an I/O framework simultaneously. Conversely, C-Coupler2.0 benefits from CIOFC1.0 to write and read restart data files in
parallel; driven by CIOFC1.0, the dynamic 3-D interpolation of C-Coupler2.0 was further improved; the output time series manager of CIOFC1.0 can help the future development of aperiodical model coupling functionality if required.

The empirical results in Section 4.3 revealed that the number of I/O processes for fastest data I/O highly depends on the field size. A component model generally inputs and outputs 2-D and 3-D fields in simulation, whereas there may be no number of
I/O processes that achieves the fastest data input and output for 3-D and 2-D fields simultaneously. For example, at the fine resolution, 15 I/O processes achieved the fastest data output for 3-D and 2-D fields, but did not achieve the fastest input of a 3-D field. As a coupled model can include different component models at different resolutions, and thus different field sizes, different component models may require different settings for the numbers of I/O processes. CIOFC1.0 will become a part of the next generation of C-Coupler, C-Coupler3, which will allow users to specify the I/O process numbers (e.g., for data input
and output of 2-D and 3-D fields respectively) separately for each component model.

Related works have already shown the effectiveness of asynchronous I/O in further reducing the I/O overhead for model simulations (Godoy et al., 2020; Yepes-Arbós et al., 2022; Huang et al., 2014). Currently, CIOFC1.0 does not implement asynchronous I/O. Our future work will seek to incorporate asynchronous I/O in C-Coupler3; we will also consider how the
various component models in a coupled model cooperatively use the asynchronous I/O functionality.

*Code availability*. The source code of CIOFC1.0 can be viewed and run with C-Coupler2 and the test model via https://doi.org/10.5281/zenodo.6345846 (Yu, 2022) (please contact us for authorization before using CIOFC1.0 for developing a system).



*Author contributions.* XY contributed to the software design, was responsible for code development, software testing and experimental evaluation of CIOFC1.0, and co-led paper writing. LL initiated this research, was responsible for the motivation and design of CIOFC1.0, supervised XY, and co-led paper writing. CS, ZZ, QJ, BZ and HY contributed to code development, software testing and evaluation. BW contributed to the motivation and evaluation. All authors contributed to improvement of ideas and paper writing.

*Competing interests.* The authors declare that they have no conflict of interest.

*Acknowledgements.* This work was jointly supported in part by the National Key Research Project of China (grant no. 2017YFC1501903).

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



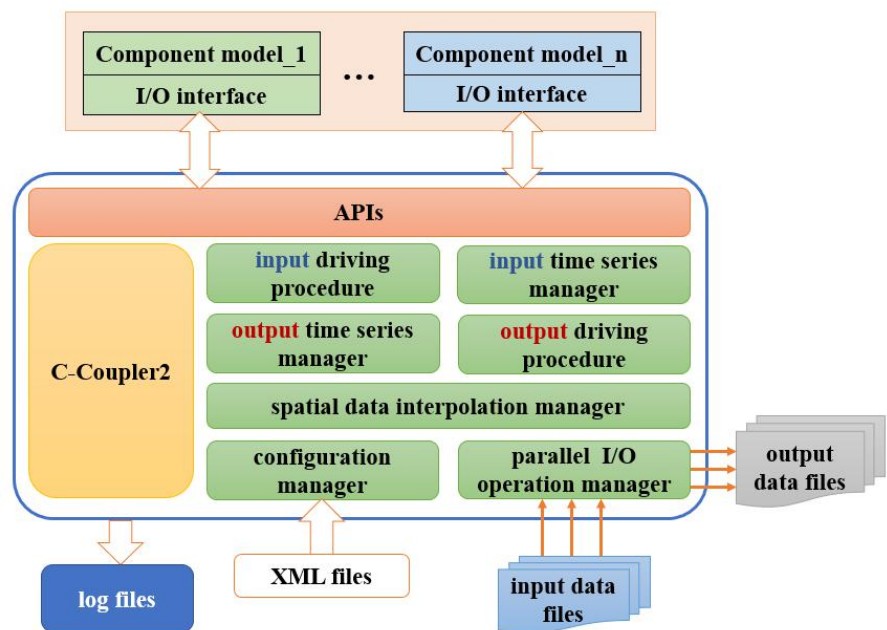

**Figure 1. Architecture of CIOFC1.0.**



```
1.  <horizontal_grids>
2.      <horizontal_grid status="on" grid_name="H2D_grid1" specification="uniform_lonlat_grid">
3.          <entry
4.              cyclic_or_acyclic="acyclic" coord_unit="degrees"
5.              num_lats="60" num_lons="60"
6.              min_lon="350.0" max_lon="20.0"    min_lat="-80.0" max_lat="80.0"
7.          />
8.      </horizontal_grid>
9.      <horizontal_grid status="on" grid_name=" H2D_grid2" specification="grid_data_file_field">
10.         <entry
11.             file_name="GEOS5-Climate/a6_nc/a6.010100.nc"
12.             edge_type="LON_LAT" coord_unit="degrees" cyclic_or_acyclic="cyclic"
13.             dim_size1="Lon-000" dim_size2="Lat-000"
14.             min_lon="0.0" max_lon="360.0" min_lat="-90.0" max_lat="90.0"
15.             center_lon="LON" center_lat="LAT"
16.         />
17.     </horizontal_grid>
18.     <horizontal_grid status="off" grid_name="H2D_grid3" specification="CCPL_grid_file">
19.         <entry
20.             file_name="GEOS5-Climate/atm_H2D_grid@atm.nc"
21.         />
22.     </horizontal_grid>
23. </horizontal_grids>
```

Figure 2. An example of specifying horizontal file grids in an XML configuration file.



```
netcdf atm_H2D_grid@atm {
dimensions:
      grid_rank = 1 ;
      grid_size = 7680 ;
variables:
      int grid_dims(grid_rank) ;
      double grid_center_lon(grid_size) ;
            grid_center_lon:long_name = "longitude" ;
            grid_center_lon:unit = "degrees" ;
      double grid_center_lat(grid_size) ;
            grid_center_lat:long_name = "latitude" ;
            grid_center_lat:unit = "degrees" ;
      int grid_imask(grid_size) ;
}
```

**Figure 3. An example of C-Coupler's default NetCDF file format for a horizontal grid.**





```
1.  <vertical_grids>
2.      <vertical_grid status="on" grid_name="V1D_ grid1" grid_type="Z"
        specification="grid_data_file_field">
3.          <entry
4.              coord_unit="hPa"
5.              coord_values="10, 20, 30, 50, 100, 125, 150, 175,200,250,300, 350, 400, 450, 500,
        550, 600, 650, 700, 750, 800, 850, 887, 925, 962, 1000"
6.          />
7.      </vertical_grid>
8.      <vertical_grid status="on" grid_name=" V1D_ grid2" grid_type="SIGMA"
        specification="uniform_vertical_grid">
9.          <entry
10.             coord_unit="hPa" order="ascending"
11.             min_value="10" max_value="1000" num_levels="20"
12.         />
13.     </vertical_grid>
14.     <vertical_grid status="on" grid_name="hybrid_grid1" grid_type="HYBRID"
15.         specification="grid_data_file_field">
16.         <entry
17.             file_name="GEOS5-Climate/GEOS-5_Reduced_Vertical_Grid_47_Levels.nc"
18.             coord_unit="hPa" top_value="1.0"
19.             coef_A="Ap" coef_B="Bp"
20.         />
21.     </vertical_grid>
22. </vertical_grids>
```

**Figure 4. An example of specifying vertical coordinates in an XML configuration file.**





```
1.   <V3D_grids>
2.       <V3D_grid status="on" grid_name="V3D_lev_grid1"
3.           mid_point_grid_name="V3D_grid1" dimension_order="v1d+h2d">
4.           <horizontal_sub_grid grid_name="H2D_grid1" />
5.           <vertical_sub_grid grid_name="V1D_ grid2">
6.               <surface_field type="external" field_name_in_file="PS" />
7.           </vertical_sub_grid>
8.       </V3D_grid>
9.       <V3D_grid status="off" grid_name="V3D_grid2">
10.          <horizontal_sub_grid grid_name="handler_output_H2D_grid" />
11.          <vertical_sub_grid grid_name="V1D_grid1" />
12.      </V3D_grid>
13.      <V3D_grid status="off" grid_name="V3D_grid3">
14.          <horizontal_sub_grid grid_name="H2D_grid1" />
15.          <vertical_sub_grid grid_name="handler_output_V1D_grid" />
16.      </V3D_grid>
17.      <V3D_grid status="on" grid_name="V3D_lev_grid4" mid_point_grid_name="V3D_grid4">
18.          <horizontal_sub_grid grid_name="H2D_grid2" />
19.          <vertical_sub_grid grid_name="V1D_grid2">
20.              <surface_field type="dynamic" field_name_in_file="GMAO-2D__PS" >
21.                  <data_time_series max_interval_unit="months" max_interval_count="1">
22.                      <data_files status="on" file_names="GEOS5-Climate/i6_nc/i6.*"
23.                          time_format_in_filename="MMDDHH" file_type="NetCDF"/>
24.                      <time_fields status="on" specification="file_name" />
25.                  </data_time_series>
26.              </surface_field>
27.          </vertical_sub_grid>
28.      </V3D_grid>
29.  </V3D_grids>
```

**Figure 5. An example of specifying 3-D grids in an XML configuration file. The horizontal sub-grids "*H2D_grid1*" and "*H2D_grid2*" and the vertical sub grids "*V1D_grid1*" and "*V1D_grid2*" have already been specified in the same configuration file**

5  **(Fig. 2 and 4).**



```
1.   <fields_output_settings>
2.       <fields_output_setting status="on" file_mid_name="F1D"
     time_format_in_data_file="YYYYMMDDSSSSS" field_specification="hybrid" default_operation="aver"
     default_float_type="real4" default_integer_type="short">
3.           <time_setting status="on" file_freq_count="2" file_freq_unit="days">
4.               <time_period status="on" freq_count="1" freq_unit="days">
5.                   …
6.               </time_period>
7.           </time_setting>
8.           <fields>
9.               
10.              
11.              
12.          </fields>
13.      </fields_output_setting>
14.      <fields_output_setting status="on" file_mid_name="F2D"
     time_format_in_data_file="YYYYMMDDSSSSS" field_specification="default">
15.          <time_setting status="on" file_freq_count="2" file_freq_unit="days" >
16.              <time_period status="on" freq_count="2" freq_unit="days" />
17.          </time_setting>
18.          <fields>
19.              
20.          </fields>
21.      </fields_output_setting>
22.  </fields_output_settings>
```

**Figure 6. An example of configurations for outputting fields in an XML configuration file. The "*V3D_grid1*" grid has already been specified in line 3 of Fig. 5.**



```
1.   <time_series_dataset name="dataset1">
2.       <data_time_series status="on">
3.           <data_files status="on" file_names="amip_bc.gamil.128x060.1951-2010.nc"
     file_type="NetCDF"/>
4.           <time_fields status="on" specification="file_field">
5.               <file_field variable="date" time_format_in_datafile="YYYYMMDD" />
6.               <file_field variable="datesec" time_format_in_datafile="SSSSS" />
7.           </time_fields>
8.       </data_time_series>
9.       <horizontal_grids>
10.          <horizontal_grid status="on" grid_name="time_field_test_H2D_grid"
     specification="grid_data_file_field">
11.              <entry
12.                  file_name="amip_bc.gamil.128x060.1951-2010.nc"
13.                  edge_type="LON_LAT" coord_unit="degrees" cyclic_or_acyclic="cyclic"
14.                  dim_size1="lon" dim_size2="lat"
15.                  min_lon="0.0" max_lon="360.0" min_lat="-90.0" max_lat="90.0"
16.                  center_lon="lon" center_lat="lat"
17.                  annotation="time field test h2d grid"
18.              />
19.          </horizontal_grid>
20.      </horizontal_grids>
21.      <input_fields>
22.          
23.      </input_fields>
24.  </time_series_dataset>
```

(a)

```
1.   <time_series_dataset name="dataset2">
2.       <data_time_series>
3.           <data_files status="on" files_name="GEOS5-Climate/a3_nc/evap_a3.*.nc"
     time_format="MMDDHH" file_type="NetCDF"/>
4.           <time_fields status="on" specification="file_name" time_point_type="start" />
5.       </data_time_series>
6.       <horizontal_grids>
7.           <horizontal_grid status="on" grid_name="H2D_grid1"
     specification="grid_data_file_field">
8.               <entry
9.                   edge_type="LON_LAT" coord_unit="degrees" cyclic_or_acyclic="cyclic"
10.                  dim_size1="Lon-000" dim_size2="Lat-000"
11.                  min_lon="0.0" max_lon="360.0"
12.                  min_lat="-90.0" max_lat="90.0"
13.                  center_lon="LON" center_lat="LAT"
14.                  annotation="register GC datamodel H2D grid"
15.              />
16.          </horizontal_grid>
17.      </horizontal_grids>
18.      <input_fields>
19.          
20.          
21.      </input_fields>
```

(b)

**Figure 7. Examples of configurations for time-series information of input data files: specifying time points through (a) the names of data files and (b) the time fields in data files.**





```
1.  <input_instances>
2.      <input_instance status="on" instance_name="Datainst_dataset1" dataset_name="dataset1">
3.          <time_mapping_configurations>
4.              <period_setting status="on" period_data_start_time="2006" period_time_format="YYYY"
    period_unit="nyears" period_count="1"  />
5.              <offset_setting status="on" offset_unit="nyears" offset_count="5"  />
6.          </time_mapping_configurations>
7.          <field_renamings>
8.              <entry name_in_model="TSKIN2" name_in_file="GMAO-2DS_TSKIN" />
9.              <entry name_in_model="EFLUX2" name_in_file="GMAO-2DS_EFLUX" />
10.         </field_renamings>
11.     </input_instance>
12. </input_instances>
```

Figure 8. An example of configuring an input instance. The dataset "dataset1" corresponds to that specified in Fig. 7(a).





1. int ncmpi_put_vara_<type> (int ncid, int varid, const MPI_Offset start[], const MPI_Offset count[], const <C type> *buf);
2. int ncmpi_put_vara    (int ncid, int varid, const MPI_Offset start[], const MPI_Offset count[], const void *buf, MPI_Offset bufcount, MPI_Datatype buftype);
3. int ncmpi_put_vara_<type>_all (int ncid, int varid, const MPI_Offset start[], const MPI_Offset count[], const <C type> *buf);
4. int ncmpi_put_vara_all (int ncid, int varid, const MPI_Offset start[], const MPI_Offset count[], const void *buf, MPI_Offset bufcount, MPI_Datatype buftype);

**Figure 9. PnetCDF APIs for writing data in parallel.**





**Figure 10. An example of two kinds of parallel decompositions. (a) Regular 2-D decomposition, with each process associated with one subdomain. (b) Round-robin-based decomposition associates each process with multiple subdomains.**





```
1.  <?xml version="1.0" ?>
2.  <Time_setting
3.      case_name="original_case"
4.      model_name="ideal_model_for_CCPL2"
5.      run_type="initial"
6.      leap_year="off"
7.      start_date="20060101"
8.      start_second="00000"
9.      reference_date="00010101"
10.     rest_freq_unit="seconds"
11.     rest_freq_count="43200"
12.     stop_option="date"
13.     stop_date="20070101"
14.     stop_second="00000"
15.     stop_n="1"
16.     max_num_pio_proc="3"
17. />
```

**Figure 11. An example of specifying the global number of I/O processes in C-Coupler's experiment setup configuration file via the parameter "max_num_pio_proc".**





```
1.  <fields_output_settings>
2.      <fields_output_setting status="on" file_mid_name="F1D" time_format_in_data_file="YYYYMMDDSSSSS"
    field_specification="hybrid" default_operation="aver" default_float_type="real4" default_integer_type="short">
3.          <time_setting status="on" file_freq_count="2" file_freq_unit="days">
4.              <time_period status="on" freq_count="1" freq_unit="days">
5.                  <time_slots status="on" time_format="hours" slots="[1 , 2],[9,10], [17, 18]" freq_count="1"
    freq_unit="hour">
6.                      <time_points status="on" time_points="1200,2400" time_format="seconds"/>
7.                  </time_slots>
8.                  <time_slots status="on" time_format="hours" slots="[5,6],[13,15],[22,22]" freq_count="2"
    freq_unit="hour">
9.                      <time_points status="on" time_points="2400,4800" time_format="seconds"/>
10.                 </time_slots>
11.             </time_period>
12.         </time_setting>
13.         <fields>
14.             
15.         </fields>
16.     </fields_output_setting>
17. </fields_output_settings>
```

(a)

```
date = 20060101, 20060101, 20060101, 20060101, 20060101, 20060101, 20060101,
    20060101, 20060101, 20060101, 20060101, 20060101, 20060101, 20060101,
    20060101, 20060101, 20060101, 20060101, 20060101 ;
datesec = 4800, 6000, 8400, 9600, 20400, 22800, 33600, 34800, 37200, 38400,
    49200, 51600, 56400, 62400, 63600, 66000, 67200, 81600 ;
```

(b)

**Figure 12. An example of specifying an output time series (a) and the corresponding time points in a day (b).**



```
%%% OUTPUT MENU %%%        : 123456789.123456789.123456789.1--1=ZERO+2=BPCH
Schedule output for JAN : 300000000000000000000000000000000
Schedule output for FEB : 300000000000000000000000000000
Schedule output for MAR : 3000000000000000000000000000000000
Schedule output for APR : 300000000000000000000000000000000
Schedule output for MAY : 3000000000000000000000000000000000
Schedule output for JUN : 3333333333333333333333333333333
Schedule output for JUL : 330000000000000000000000000000000
Schedule output for AUG : 3000000000000000000000000000000000
Schedule output for SEP : 300000000000000000000000000000000
Schedule output for OCT : 3000000000000000000000000000000000
Schedule output for NOV : 300000000000000000000000000000000
Schedule output for DEC : 3000000000000000000000000000000000
```

**Figure 13. An example of output setting of GEOS-Chem. A number "3" means that GEOS-Chem outputs fields every three hours on the corresponding date of every year, while a number "0" means that GEOS-Chem does not output any field on the corresponding date of every year.**





```
1.  <time_setting status="on">
2.      <time_period status="on" freq_count="1" freq_unit="years">
3.          <time_slots status="on" time_format="MM"    slots="[1,5],[8,12]" freq_count="1" freq_unit="months">
4.              <time_slots status="on" time_format="DD" slots="[1,1]" freq_count="3" freq_unit="hours"/>
5.          </time_slots>
6.          <time_slots status="on" time_format="MM"    slots="[6,6]" freq_count="3" freq_unit="hours" />
7.          <time_slots status="on" time_format="MM"    slots="[7,7]" freq_count="1" freq_unit="months" >
8.              <time_slots status="on" time_format="DD" slots="[1,2]" freq_count="3" freq_unit="hours"/>
9.          </time_slots>
10.     </time_period>
11. </time_setting>
```

**Figure 14. An example of the specification of the output time series for GEOS-Chem in CIOFC1.0 corresponding to the specification in Fig. 13.**



```
1.  <time_setting status="on">
2.      <time_slots status="on" time_format="ndays"   slots="[0, 4]" freq_count="3" freq_unit="hours" />
3.      <time_slots status="on" time_format="ndays"   slots="[5, 30]" freq_count="6" freq_unit="hours" />
4.  </time_setting>
```

**Figure 15. An example of the specification of the output time series of GRAPES in CIOFC1.0.**





```
1.   <time_setting status="on">
2.       <time_period status="on" freq_count="1" freq_unit="hour">
3.           <time_slots status="on" time_format="days" slots="[1,2], [9,10], [17,18]" freq_count="1"
     freq_unit="hour"/>
4.       </time_period>
5.   </time_setting>
```

**Figure 16. An example of an incorrect output time series setting. The frequency unit in the outer level of the specification should be no smaller than that in the inner level.**





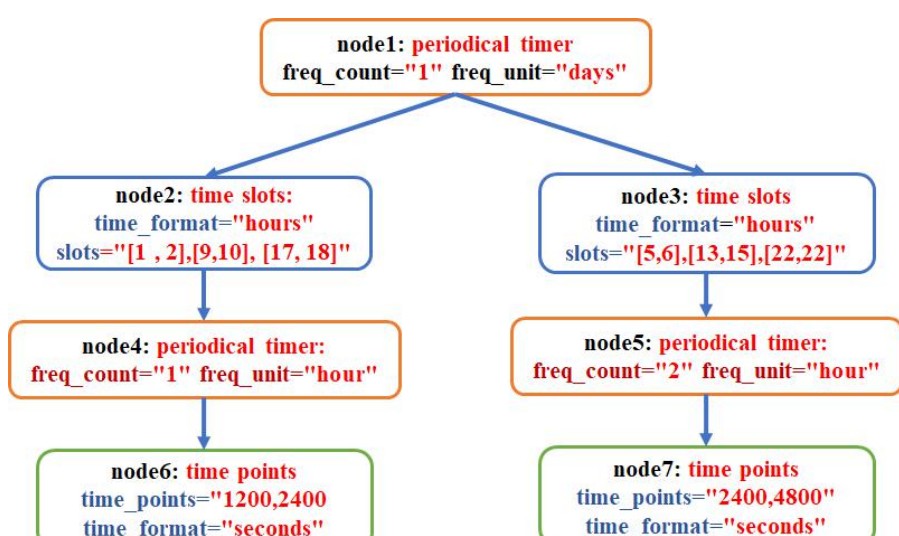

**Figure 17. A visualization of the timer tree corresponding to the configuration in Fig. 12.**





```
1.  integer FUNCTION CCPL_register_configurable_output_handler(...)
2.  implicit none
3.  character(len=*), intent(in)                          :: handler_name
4.  integer,          intent(in)                          :: num_field_instances
5.  integer,          intent(in), dimension(:)            :: field_instance_ids
6.  character(len=*), intent(in)                          :: configuration_name
7.  logical,          intent(in)                          :: implicit_or_explicit
8.  integer,          intent(in), optional               :: output_grid_id
9.  integer,          intent(in), optional               :: handler_output_h2d_grid_id
10. integer,          intent(in), optional               :: handler_output_v1d_grid_id
11. integer,          intent(in), optional               :: sampling_timer_id
12. character(len=*), intent(in), optional               :: annotation
```

**Figure 18. The argument list of the API *CCPL_register_configurable_output_handler*.**



```
1.   integer FUNCTION CCPL_register_normal_output_handler(...)
2.   implicit none
3.   integer,            intent(in)                          :: num_field_instances
4.   integer,            intent(in), dimension(:)            :: field_instance_ids
5.   character(len=*), intent(in)                            :: file_name
6.   character(len=*), intent(in)                            :: file_type
7.   logical,            intent(in)                          :: implicit_or_explicit
8.   integer,            intent(in), optional               :: sampling_timer_id
9.   integer,            intent(in), optional               :: output_timer_id
10.  integer,            intent(in), optional               :: file_timer_id
11.  integer,            intent(in), optional               :: output_grid_id
12.  integer,            intent(in), optional               :: inst_or_aver
13.  character(len=*),   intent(in), optional               :: float_datatype
14.  character(len=*),   intent(in), optional               :: integer_datatype
15.  character(len=*),   intent(in), optional               :: annotation
```

**Figure 19. The argument list of the API *CCPL_register_normal_output_handler*.**



```
1.   logical FUNCTION CCPL_handle_normal_explicit_output(handler_id, bypass_timer, annotation)
2.   implicit none
3.   integer,          intent(in)                      :: handler_id
4.   logical,          intent(in), optional            :: bypass_timer
5.   character(len=*), intent(in), optional            :: annotation
```

**Figure 20. The argument list of the API *CCPL_handle_normal_explicit_output*.**



```
1.  <time_series_dataset name="dataset_example">
2.     <time_series_fields enable_extrapolate="on">
3.        <data_files status="on" file_names="file_name.*.nc" time_format= "YYYYMMDD"
   file_type="NetCDF"/>
4.        <time_fields status="on" specification="file_field" time_point_type="middle" >
5.           <file_field variable="date" time_format_in_datafile="YYYYMMDD" />
6.        </time_fields>
7.        <input_fields>
8.           
9.        </input_fields>
10.    </time_series_fields>
11. </time_series_dataset>
```

**Figure 21. An example configuration of a daily input dataset, where the time series is specified via the corresponding time field.**

**The type of time points is set as "*middle*", meaning that the time point represents the 43200th second of a day.**





```
1.  <input_instances>
2.      <input_instance status="on" instance_name="Datainst_6hours"
    dataset_name="dataset_example">
3.          <time_mapping_configurations>
4.              <period_setting status="on" period_data_start_time="20050101"
    period_time_format="YYYYMMDD" period_unit="ndays" period_count="20"   />
5.                  <offset_setting status="on" offset_unit="ndays" offset_count="0"   />
6.          </time_mapping_configurations>
7.          <field_renamings>
8.              <entry name_in_model="tskin_in" name_in_file="GMAO-2DS__TSKIN" />
9.          </field_renamings>
10.      </input_instance>
11. </input_instances>
```

**Figure 22. An example configuration of time mapping rule corresponding to the configuration in Fig. 21. and Table 2.**





```
1.  integer FUNCTION CCPL_register_input_handler(...)
2.  implicit none
3.  character(len=*), intent(in)                          :: handler_name
4.  integer,          intent(in), dimension(:)            :: field_instances_ids
5.  integer,          intent(in)                          :: num_fields
6.  character(len=*), intent(in)                          :: config_input_instance_name
7.  integer,          intent(in)                          :: input_timer_id
8.  logical,          intent(in), dimension(:), target, optional  :: necessity
9.  logical,          intent(in), dimension(:), target, optional  :: field_connected_status
10. character(len=*),        intent(in), optional         :: annotation
```

**Figure 23. The argument list of the API *CCPL_register_input_handler*.**





```
1.  logical FUNCTION CCPL_execute_input_handler(handler_id, annotation)
2.
3.  implicit none
4.  integer,            intent(in)                  :: handler_id
5.  character(len=*), intent(in), optional          :: annotation
```

**Figure 24. The argument list of the API *CCPL_execute_input_handler*.**





```
1.  logical FUNCTION CCPL_readin_field_from_dataFile(...)
2.  implicit none
3.  integer,           intent(in)                        :: field_instance_id
4.  character(len=*),  intent(in)                        :: data_file_name
5.  character(len=*),  intent(in)                        :: file_type
6.  character(len=*),  intent(in), optional              :: field_name_in_file
7.  logical,           intent(in), optional              :: necessity
8.  integer,           intent(in), optional              :: grid_id_for_file
9.  character(len=*),  intent(in), optional              :: annotation
```

**Figure 25. The argument list of the API *CCPL_readin_field_from_dataFile*.**



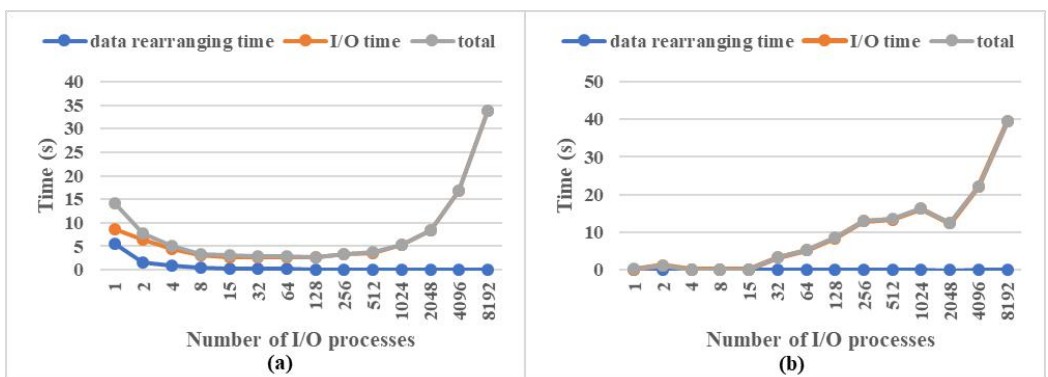

Figure 26. Time required for data rearranging, I/O, and the total time for outputting (a) a 3-D field and (b) a 2-D field at fine

resolution. 10,048 processes were used to run the test model.



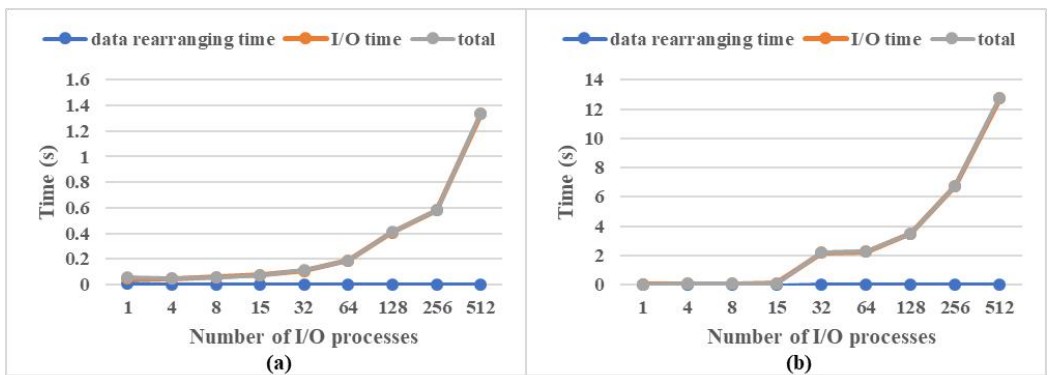

**Figure 27. As Fig. 26, but for coarse resolution. 512 processes were used to run the test model.**





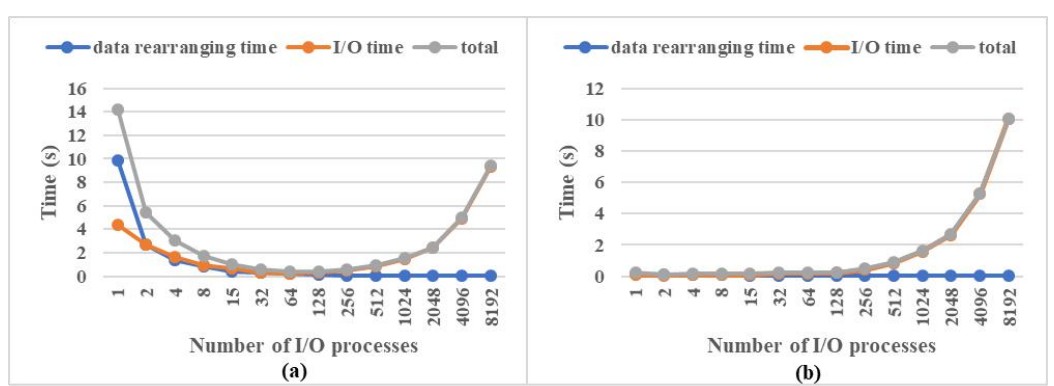

**Figure 28. As Fig. 26, but for inputting a field.**





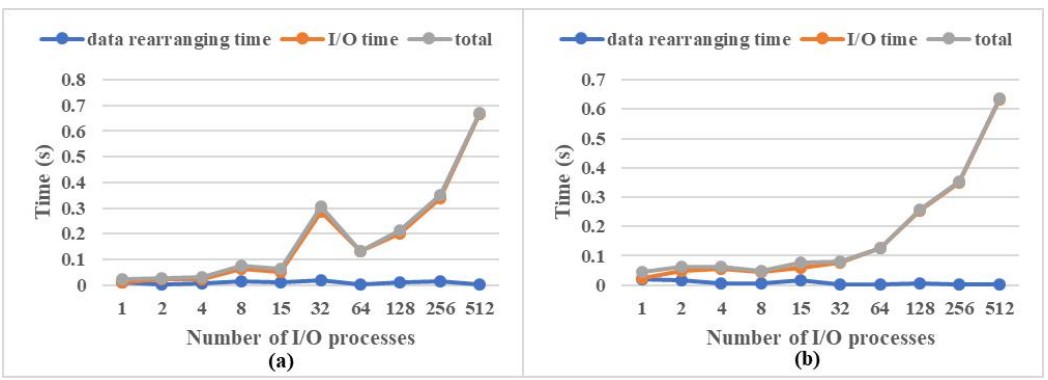

**Figure 29. As Fig. 27, but for inputting a field.**





**Table 1. Examples of the time formats supported in CIOFC1.0.**

| Time formats | Description |
|---|---|
| SSSSS | The time of day in seconds (one day has 86,400 s) |
| MMDD, MM-DD | A calendar date given as month and day |
| YYYY | The calendar year of a date |
| YYYYMMDD, YYYY-MM-DD | The calendar year (YYYY), month (MM), and day (DD) of a date |
| HHMMSS, HH-MM-SS, HH:MM:SS | The time of day giving the hour (HH), minute (MM), and second (SS) |
| MMDD-HHMM, MM-DD.HH-MM, MM-DD-HH-MM, MMDDHHMM | A calendar date given as month, day, hour, and minute |
| MMDDHH, MMDD.HH, MMDD-HH, MM-DD.HH, MM-DD-HH | A calendar date and hour of day combined |
| YYYYMMDD.SSSSS, YYYY-MM-DD.SSSSS, YYYYMMDD-SSSSS, YYYY-MM-DD-SSSSS, YYYYMMDDSSSSS | A calendar date with the time of day (in seconds) combined |
| nyears, years, nyear, year | The number of simulation years from the start |
| nseconds, seconds, second, nsecond | The number of simulation seconds from the start |
| nsteps, steps, nstep, step | The number of simulation steps from the start |





**Table 2. An example of managing time points in an input dataset with four data files corresponding to the configuration in Fig. 21.**

**(a) Time fields in each data file. (b) The information of time points recorded by CIOFC1.0.**

| file name | file_name.20050101.nc | file_name.20050106.nc | file_name.20050111.nc | file_name.20050116.nc |
|-----------|-----------------------|-----------------------|-----------------------|-----------------------|
| time field | 20050101, 20050102 20050103, 20050104 20050105 | 20050106, 20050107 20050108, 20050109 20050110 | 20050111, 20050112 20050113, 20050114 20050115 | 20050116, 20050117 20050118, 20050119 20050120 |

(a)

| time points | file name | file index | time points | file name | file index |
|-------------|-----------|------------|-------------|-----------|------------|
| 20150101-43200 | file_name.20050101.nc | 0 | 20150111-43200 | file_name.20050111.nc | 2 |
| 20150102-43200 | file_name.20050101.nc | 0 | 20150112-43200 | file_name.20050111.nc | 2 |
| 20150103-43200 | file_name.20050101.nc | 0 | 20150113-43200 | file_name.20050111.nc | 2 |
| 20150104-43200 | file_name.20050101.nc | 0 | 20150114-43200 | file_name.20050111.nc | 2 |
| 20150105-43200 | file_name.20050101.nc | 0 | 20150115-43200 | file_name.20050111.nc | 2 |
| 20150106-43200 | file_name.20050106.nc | 1 | 20150116-43200 | file_name.20050116.nc | 3 |
| 20150107-43200 | file_name.20050106.nc | 1 | 20150117-43200 | file_name.20050116.nc | 3 |
| 20150108-43200 | file_name.20050106.nc | 1 | 20150118-43200 | file_name.20050116.nc | 3 |
| 20150109-43200 | file_name.20050106.nc | 1 | 20150119-43200 | file_name.20050116.nc | 3 |
| 20150110-43200 | file_name.20050106.nc | 1 | 20150120-43200 | file_name.20050116.nc | 3 |

(b)