# Peer review of "CIOFC1.0: a Common Parallel Input/Output Framework Based on C-Coupler2.0"

_Geoscientific Model Development, 2022_

## Author Comment (AC3)

We thank the anonymous reviewers for carefully reading the manuscript and for providing the very valuable comments. The review comments reveal weak points of this manuscript and give us a lot of suggestions for further revision. Guided by the review comments, we will try to significantly improve the manuscript in the following main aspects:

1. We will improve the evaluation of CIOFC1.0 by including the discussions and tests regarding with other I/O frameworks that are frequently used by other major climate models. We will include the performance test result of a real climate model besides the test model we are currently using and make more explanations about how CIOFC can be used by other component models.

2. We will improve the introduction of the parallel I/O manager and add the reference about prior work when discussing I/O decompositions, data rearrangements and grouping I/O processes as a subset of the model compute processes.

3. We will improve the introduction of the output time series manager and the spatial data interpolation manager for the comparison with the implementations of timers and the performance of interpolation function in other major climate models.

4. Regarding with the reproducibility of the test model, we will significantly modify the *Test Model with C-Coupler2 with CIOFC User's Guide* with more details for running the test model. Some modifications need to be made for running the test model on a new platform or under a new directory, but we haven't made these details clear in the user's guide.

---

## Author Response (AR1)

Dear Reviewer,

We thank Reviewer #1 for the comments and suggestions. We have modified the manuscript accordingly. In the following, we will reply them one by one.

1. Despite my efforts, it was not possible to compile the test case provided with the source code. The reason is certainly a lack of time that would have been necessary to fully investigate the issues. However, it seems to me that the configuration procedure is rather complex, fitted to the machine's architecture of the C-Coupler community and, even though only a small number of external library are required (NetCDF, pnetCDF and MPI), it is difficult to exactly identify the origin of the problem. Sadly, this is a problem that can commonly happen in our community. A simple "makefile" with few compiling option parameters would have been much more convenient …

**Response:** We modified the user's guide for running the test model with CIOFC1.0. We are sorry that efforts for installing the NetCDF, pnetCDF and MPI are necessary.

2. More details would be required to have a better picture of the CIOFC compatibility with a non "C-Coupled" model, e.g. (i) is the tool able to handle masks (sparse matrix) ? or (ii) what are the available interpolations and are they conservative ?

**Response:** CIOFC was implemented based on the functions C-Coupler already has and inherits all the spatial interpolation features in C-Coupler2, so that CIOFC is able to handle masks during input and output. Please refer to Section 3.2 (P8L4~P8L9) and the Section 5 (P22L28~P23L6) of the revised manuscript.

3. One would have like to find there a larger set of performance measurements, not only for competition spirit, but also to be able to evaluate the limits of the chosen technology. However, these limits (synchronicity, single server for the whole ESM components) are mentioned in conclusion, which suggests future fruitful enhancements.

**Response:** More results about the comparison with other I/O framework (XIOS2.0) and the performance in a real model (MCV) have been added. Please refer to Section 4.3 (P22L6~P22L17) of the revised manuscript.

4. the XML format is adopted to select the data that have to be transferred. Even though this a standard choice in the community, I wonder if the XML parser choice and its maintenance could become a problem during the development and would like to know the authors feeling about that.

**Response:** An XML parser, TinyXML, has been included in C-Coupler2.0. We find that existing functionalities of TinyXML

are enough for C-Coupler.

5. Are the output CMOR or CF compliant and if not, why?

**Response:** Currently, CIOFC1.0 is not CMOR or CF compliant. When CIOFC1.0 output model fields, attributes (e.g., field name, units, and long name) of each field will also be written into an output file. Field attributes can be conveniently specified in the corresponding XML configuration files of CIOFC1.0 and C-Coupler2.0 for different purposes (please refer to Section 5 (P23L9~P23L12) of the revised manuscript). We believe that CIOFC1.0 can be made CMOR or CF compliant in the future.

6. Concerning the launching of the I/O server processes, it is precised that they are considered as a « subset of model processes (p9, l4)». Their number is an XML file parameter (max_num_pio_proc) but it is certainly also necessary to increase the number of a model process number accordingly? If yes, on which model should the user do that ? How are these I/O processes identified by the model(s) at initialisation to avoid including them in the pool of its compute processes ?

**Response:** More details of the implementation about "*max_num_pio_proc*" are included in the revised manuscript. Please refer to Section 3.3 (P10L5~P10L7). Currently, a simple strategy is used in CIOFC1.0 for evenly picking out I/O processes from model processes according to the model process IDs. Please refer to P23L27 to P24L2.

7. The procedure which aims to select data from file or model is precisely described, sometimes with too much verbosity, e.g. the one related the output time serie manager (§3.4). It could also be interesting to describe how this information is transmitted to the library (XML parser). How the input/output data of the model is updated could also be unclear to the reader, since no model array to be updated/transferred are provided via the CIOFC API (for writing in output mode, or update in input mode). A check to the example source code shows that it is done via the C-Coupler API, but it could be interesting to mention it in the article.

Response: The manuscript has been modified according to the above questions. Please refer to Section 3.5 (P12L11~P12L16).

8. I also wonder if a CIOFC output or input handler can be set during runtime, or differently said, if the way the model data are modified by input (or the output files) can be changed interactively during simulation ? If yes, can the interpolation be

recomputed and if yes, how long does it take (performance)?

**Response:** currently, the way of data input/output of a handler cannot be changed after the handler has been initialized.

9. The authors emphasize the « flexibility to change the source of each boundary field » (p15, l17 ), but can a variable be switched during runtime from values read in file (input) and read from a model (coupling) ? In another possible configuration, can these two sources contribute simultaneously to define a mixed variable, following a geographical mask (e.g. SST coupled from a model, and lake temperature read in a file?)

**Response:** Source of a boundary field is determined when initializing the model as well as an input handler, while cannot be changed during runtime. A possible solution for the above requirement is to maintain two sources (one source from input files and one source from model coupling) of a variable, and to determine how to use these two sources during runtime (e.g., select one of them, or merge them with masks or fractions).

10. For readers interested by comparison (since absolute values are provided in Fig 26-29), it would be good to give more details about the experimental protocol leading to the computing performance measurement at §4.3. For example, the file system kind (and its theoretical performance) could be mentioned. From the model part, the output frequency during the measurement must also be provided.

**Response:** More details about the experiments have been provided. Please refer to Section 4 (P18L16~P18L17) and Section 4.3 (P20L19~P20L21) of the revised manuscript.

11. During measures, the « numerical calculations in real models are neglected» (p18, l3), but are you sure that no interaction between computations and CIOFC mapping could occur? Other interaction would be interesting to avoid: other users could stress the file system during the test. Was the machine empty during the measurements? In addition, do you think that three measurements per each test case are enough to neglect the variability induced by the perturbation mentioned above?

**Response:** We've added the process of spatial interpolation during input/output in the test model (P21L28~P22L4). We've also added the performance test of CIOFC1.0 in an atmospheric model MCV. Please refer to Section 4.3 (P22L6~P22L12) of the revised manuscript. We also noted that applications of other users running on the same machine could stress the file system. We therefore always conducted tests at midnights, in order to get stable test results.

12. If the output frequency is kept constant, how this number was chosen and does its value change the results? If yes, how? This could be interesting to understand how much asynchronous output (or buffering?) is needed. Could you precise what is the kind of MPI communications in C-Coupler between models and I/O servers (MPI_Send, MPI_Bsend, MPI_Isend?)

**Response:** The output frequency (or output time series) is determined by users via the corresponding configurations. We think that it is case dependent about how much asynchronous output is needed. We find that the output frequency does not significantly impact the performance of outputting/inputting (please refer to Fig. 29 of the revised manuscript). MPI_Isend and MPI_Irecv are used for data transferring among models processes and I/O processes.

Dear Reviewer,

We thank Reviewer #2 for the comments and suggestions. We have modified the manuscript accordingly. In the following, we will reply them one by one.

1. The paper would significantly benefit by including discussions and comparisons with I/O frameworks used by other major climate models. Also, including performance of a climate model component instead of a test model and comparing it with other I/O frameworks would be useful to the reader.

**Response**: We've added the performance test of CIOFC1.0 in atmospheric model MCV. Please refer to Section 4.3 (P22L6~P22L12) of the revised manuscript.

2. The paper includes a brief survey of the existing coupler software and frameworks, however it would be useful to include more references to I/O frameworks and libraries (e.g. the NetCDF library which has Parallel I/O support, the PIO library used by CESM and other climate models, the SCORPIO library used by the E3SM climate model, the I/O libraries used by frameworks like ESMF, the CFIO library) used by other climate models. More discussions comparing the work described in this paper with these I/O frameworks (CFIO, XIOS, PIO, SCORPIO, NetCDF, ESMF etc) would help in understanding the original contributions in this paper and add to the motivation for this work.

**Response**: More discussions about other I/O software or frameworks have been added in Section 1 (P2L9~P2L20) of the revised manuscript.

3. Although the paper mentions (p2, l20) that CIOFC1.0 can be used by other component models (apart from the community coupler) it is not apparent from the paper how it can be achieved, especially if the component model does not use C-Coupler2.0. From the provided source code and discussions in the paper the CIOFC1.0 framework is not a separate library (is part of the C-Coupler2.0 library), so integrating it with a separate component model (GAMIL, GEOS, GRAPES) would demonstrate how it can be used by component models in an earth system model.

**Response**: CIOFC1.0 has been integrated in a real model MCV. Please refer to Section 4.3 (P22L6~P22L12) and Section 5 (P22L28~P23L6) of the revised manuscript.

4. When discussing the I/O configuration manager that uses XML-formatted inputs (Section 3.1) it would be useful to

compare the approach here with other climate models that handle structured and unstructured grids (and have similar issues to deal with - support different types of grids, vertical coordinates etc).

**Response**: We've added the comparison of the existing work with our implementation. Please refer to Section 3.1.1 (P4L22~P5L4) of the revised manuscript.

5. The spatial interpolation manager (Section 3.2) in CIOFC1.0 uses the interpolation algorithms from the coupler so having the functionality in the I/O framework is useful when integrating model components directly with CIOFC1.0 (if not, can't this functionality be moved inside the coupler?). So showing integration of the I/O framework with a model component (that does not use the C-Coupler2.0) would have been useful here.

**Response:** CIOFC1.0 is implemented on top of C-Coupler2.0, so that CIOFC1.0 inherits the interpolation algorithms in C-Coupler2.0. When a model uses CIOFC1.0, C-Coupler2.0 is also used at the same time. Integration of the I/O framework with a model component that does not use the C-Coupler2.0 has been discussed in Section 5 (P22L28~P23L6) of the revised manuscript.

6. When implementing the Parallel I/O operation (Section 3.3) were there any discussions on supporting low level libraries other than PnetCDF or using formats other than NetCDF? Were there any s/w design decisions made based on adding support for new libraries or formats in the future?

**Response:** CIOFC1.0 only supports the classic NetCDF file format (e.g., serial I/O with NetCDF and parallel I/O with PnetCDF) currently. More data file formats such as NetCDF4 and GRIB will be supported in the near future (P23L12~P23L14).

7. In Section 3.3 (p9, l5) when discussing I/O decompositions, data rearrangements and grouping I/O processes as a subset of the model compute processes it would be useful to refer and discuss prior work in this area (specifically work done by J Dennis et al on the Parallel I/O library). Also it would be informative to discuss how the framework handles reading and writing multiple model variables into multiple files, since model components write 100s of variables in a typical simulation run.

**Response**: Discussions about the prior works have been added into the revised manuscript. Please refer to Section 3.3

(P9L8~P9L12).

8. The implementation of a recursive tree based timer is discussed in Section 3.4 (p11, l10). It would be useful to compare the approach here with current implementations of timers in other major climate models. Typically timers are useful in climate models in the model driver and other model components, why did the authors choose to implement the recursive tree based timer in the I/O framework instead of the C-Coupler2.0 (p11, l1)?

**Response:** The periodical timer in C-Coupler2.0 is unable to support the complex time series as used in GEOS-Chem (different period on different calendar dates) or GRAPES (different period in different simulation segments). Please refer to Section 3.4 (P10L10~P10L15) of the revised manuscript.

9. The implementation of the output driving procedure is discussed in Section 3.5 (p11, l25). Again it would be useful to compare the approach here with approaches used by other coupler software/frameworks like MCT, NUOPC etc.

**Response**: Some comparison with XIOS has been added into Section 3.5 (P12L11~P12L16) of the revised manuscript.

10. In Section 4 the evaluation of the I/O framework was performed using a test model (p17, l19)

1) It would have also been useful to compare the serial (C-Coupler2.0 without CIOFC1.0) vs parallel (CIOFC1.0) I/O for a model component run (e.g. GRAPES, GEOS, GAMIL).

**Response**: The performance with a real model (MCV) has been added in Section 4.3 (P22L6~P22L12) of the revised manuscript.

2) When evaluating performance (p19, l23) a fine resolution vs coarse resolution is used for the variables being written out, however it is not apparent whether the data is on a structured or unstructured grid (The test model does seem to support both grids - p18, l4). (For example the performance, especially the data rearrangement time, would vary significantly depending on whether the data is on a structured grid with regular 2d decomposition or on an unstructured cubed sphere grid)

**Response**: The performance results in Fig. 25 to Fig. 28 use longitude-latitude model grids and the same file grids. We've modified the experiment description in Section 4.3 (P20L15~P20L21). Moreover, the data rearranging performance under

different types of parallel decompositions have also been evaluated in Section 4.3 (P21L28~P22L4) of the revised manuscript.

3) It would be useful to see the average model I/O write/read throughput with the I/O framework (This would include all I/O costs - creating files, defining I/O decompositions, data rearrangement, filesystem write time etc & would also include 100s of multi dimensional model variables written out in a typical run). The I/O throughput for a single variable is sometimes not representative of the overall I/O throughput.

**Response**: The corresponding result and discussion have been added in Section 4.3 (P20L24~P20L27, P21L24~P21L26) of the revised manuscript.

4) Including comparison of the I/O throughput with other I/O libraries/frameworks (CFIO, XIOS, PIO, SCORPIO, NetCDF, ESMF etc) would be useful here.

**Response**: Comparison with XIOS has been added in Section 4.3 (P22L14~P22L17) of the revised manuscript.

5) In Section 4.3 (p20, l1-10) the authors discuss the difference in I/O write/read performance for the different number of I/O processes. It would have been useful to include discussion on how this information helps in chosing the I/O framework configuration parameters (e.g. number of I/O processes chosen) for a model run that outputs many 1D, 2D & 3D variables

**Response**: We've added the corresponding discussion of the performance conclusion in Section 5 (P23L23~L23L27) of the revised manuscript.

6) Was there any impact on I/O performance due to the placement of the I/O processes (is it configurable by the user)? Were there any interesting aspects of the machine architecture that impacted the I/O performance (e.g. In Earthlab for example the 6D TORUS network, SSDs, fast vs ordinary storage pool etc)

**Response:** We've added the corresponding discussion in Section 5 (P23L27~P24L2) of the revised manuscript.

7) Although data interpolation is handled by the coupler, it would be useful to include some brief performance statistics that includes the performance of the data interpolation and comparison with similar frameworks (online vs offline interpolation, algorithm characteristics like conservativeness, performance of interpolation using other libs/frameworks like ESMF, MOAB etc)

**Response:** The throughput of post-processing with data interpolation has been evaluated. It is much larger than throughput of filesystem write/read (please refer to Section 4.3 (P21L28~P22L4)). C-Coupler2.0 uses offline remapping weight files in most cases to avoid the corresponding cost in model run. When an offline remapping weight file is not provided, C-Coupler2.0 will generate a remapping weight file that will be further used as an offline file in next runs.

8) As a general comment, the paper includes very detailed information on the implementation of the different software managers in the framework and some of this information can be summarized without impacting the overall quality of the paper. Similarly in the case of figures the authors could also consider showing only the relevant parts of the XML configurations and removing the list of PnetCDF APIs.

**Response**: We've made modification in the revised manuscript. Please refer to Section 3.3 (P9L5) of the revised manuscript.

11. In the conclusion section (Section 5, p21 l24) the authors mention that they intent to support asynchronous I/O in the next version of the community coupler (C-Coupler3). Did you add this flexibility into the current design of the I/O framework? How much of the I/O framework needs to change to incorporate asynchronous I/O?

**Response:** The I/O framework now is a part of C-Coupler3 while it does not support asynchronous I/O. Implementation of asynchronous I/O is our future work based on C-Coupler3. Sincerely, we do not have detailed plans of the implementation currently.

---

## Author Response (AR2)

Dear Reviewer,

We sincerely thank Reviewer #2 for the comments and suggestions. We have modified the manuscript accordingly. In the following, we will reply them one by one.

1. It would be useful to the reader to compare the XML configurations, the recursive timers used by CIOFC1.0 with other major climate models (are they compatible? any existing issues in other models that were fixed by the framework?)

**Response:** Sincerely, no major climate models has used CIOFC1.0 for improvement in data input/output. We find that, advantages of CIOFC1.0 may be not attractive enough to major climate models that already have codes for data input/output and many users, while the advantages would be more attractive to newly developed models. CIOFC1.0 has been used by a new model (i.e., MCV), which was discussed and summarized in Section 5.

2. "supports three kinds of vertical coordinates: Z, SIGMA, and HYBRID coordinates" (p5, l25) - missing references for the different kinds of coordinates?

**Response:** We've added the references in the text. Please refer to Section 3.1.1.2 (P5L25) of the revised manuscript.

3. Authors might want to rephrase "Parallelization by CIOFC1.0 can accelerate ..." (p1, l20) to "The CIOFC1.0 framework can accelerate... by parallelizing..." (specifying what was parallelized)

**Response:** We've rephrased the expression accordingly. Please refer to the Abstract (P1L20~P1L21) of the revised manuscript.

4. "This paper focus on C-Coupler..." (p2, l7) to "This paper focuses on C-Coupler..."

**Response:** We've modified the expression accordingly. Please refer to Section 1 (P2L7) of the revised manuscript.

5. "As models are under the development of finer grid resolutions, ..." (p2, l9) to "As models are using finer grid resolutions..."

**Response:** We've rephrased the expression accordingly. Please refer to Section 1 (P2L10) of the revised manuscript.

6. "instead use sequential I/O for high-resolution integration..." (p2, l21) to "instead use sequential I/O for high-resolution model output"

**Response:** We've rephrased the expression accordingly. Please refer to Section 1 (P2L22~P2L23) of the revised manuscript.

7. "the atmospheric model of FV3 (Finite Volume Cubed Sphere Finite Volume Cubed Sphere) ..." (p3, l16) to "the atmospheric model of FV3 (Finite Volume Cubed Sphere) ..."

**Response:** We've rephrased the expression accordingly. Please refer to Section 2 (P3L16~P3L17) of the revised manuscript.

8. "combing data interpolation and field input/output..." (p4, l25) to "combining data interpolation and field input/output..."

**Response:** We've rephrased the expression accordingly. Please refer to Section 3.1.1 (P4L25) of the revised manuscript.

9. "A technical question for such a question is how to specify file grids..." (p4, l25) to "A challenge ..."

**Response:** We've rephrased the expression accordingly. Please refer to Section 3.1.1 (P5L1) of the revised manuscript.

10. "We also make the I/O configuration manager provide XML configurations..." (p5, l3) to "We also implemented the I/O configuration manager to output XML configurations..."

**Response:** We've rephrased the expression accordingly. Please refer to Section 3.1.1 (P5L3~P5L4) of the revised manuscript.

11. "3.3 Implementation of the parallel I/O operation " (p8) to "3.3 Implementation of parallel I/O"

**Response:** We've rephrased the expression accordingly. Please refer to Section 3.3 (P8L24) of the revised manuscript.

12. "A fine resolution () and a coarse resolution () were used. " (p20 l15) to "A fine resolution () and a coarse resolution () fields were used for testing..."

**Response:** We've rephrased the expression accordingly. Please refer to Section 4.3 (P20L15~P20L17) of the revised manuscript.

13. "We first evaluated the performance of data output. " (p20, l23) to "We first evaluated the [framework output | model write | model output] performance..."

**Response:** We've rephrased the expression accordingly. Please refer to Section 4.3 (P20L23) of the revised manuscript.

14. "This is because the output of multiple fields can share the same file creation and..." (p20, l26) to "This is because multiple fields can be written out to the same file, sharing the file creation time,..."

**Response:** We've rephrased the expression accordingly. Please refer to Section 4.3 (P20L26~P20L27) of the revised manuscript.

15. "especially when the number of I/O processes is big..." (p21, l4) to "especially for large number of I/O processes..."

**Response:** We've rephrased the expression accordingly. Please refer to Section 4.3 (P21L3) of the revised manuscript.

16. "could be achieved " (p21) to "was achieved"

**Response:** We've rephrased the expression accordingly. Please refer to Section 4.3 (P21L8, P21L10, P2115, P21L17).

Additional modifications:

17. We added the grant no. of two research projects in the *Acknowledgements* part. These two projects essentially supported this research work. We are sorry that we forgot to acknowledged them in the previous versions of this manuscript.